# Nettle Cultivation Practices—From Open Field to Modern Hydroponics: A Case Study of Specialized Metabolites

**DOI:** 10.3390/plants11040483

**Published:** 2022-02-11

**Authors:** Nevena Opačić, Sanja Radman, Sanja Fabek Uher, Božidar Benko, Sandra Voća, Jana Šic Žlabur

**Affiliations:** 1Department of Vegetable Crops, University of Zagreb Faculty of Agriculture, Svetošimunska Cesta 25, 10000 Zagreb, Croatia; nopacic@agr.hr (N.O.); sfabek@agr.hr (S.F.U.); bbenko@agr.hr (B.B.); 2Department of Agricultural Technology, Storage and Transport, University of Zagreb Faculty of Agriculture, Svetošimunska Cesta 25, 10000 Zagreb, Croatia; svoca@agr.hr (S.V.); jszlabur@agr.hr (J.Š.Ž.)

**Keywords:** *Urtica dioica*, soilless systems, bioactive compounds, cultivated nettle, stress factors, functional properties

## Abstract

Conventional agricultural production faces numerous challenges due to the pronounced effects of climate change, particularly global warming, and drought more than ever before in history, with the primary concern being to produce adequate yields and high-quality, nutritious plant material. Likewise, people are increasingly looking for new sources of food and are becoming aware of the importance of a varied diet and its connection to health. In this sense, stinging nettle (*Urtica dioica* L.) stands out as a valuable species that is neglected as a food source, as it has a significant content of specialized metabolites, and thus has an extremely high potential for use both nutritionally and pharmacologically, but is still traditionally collected from natural habitats, so it can be of questionable quality and undefined chemical composition. Therefore, sustainable agricultural practices are increasingly shifting to modern hydroponic cultivation methods in greenhouses. The advantage lies in the easier management and control of a number of factors during cultivation (air temperature and relative humidity, balanced and rational fertilization, minimization of nitrate uptake, etc.), ensuring better conditions for the growth and development of nettle according to its needs. The aim of this review is to give an overview of the technology of stinging nettle cultivation in the field and to show the possibilities of cultivation with modern hydroponic techniques to obtain a final product of consistent and uniform quality, high content of specialized metabolites and significant nutritional value. Research on this topic is still sparse but will certainly increase in the future. Therefore, this review provides all the necessary data for such future studies.

## 1. Introduction

The current trend among consumers is to pay more attention to a balanced diet and the consumption of wild leafy plants, as they are recognized as a source of a high content of phytonutrients [1,2,3,4], which play an important role in the prevention of many chronic diseases [5]. One of these wild edible species is stinging nettle (*Urtica dioica* L.), which has been known since ancient times for its medicinal value as a herbal remedy for the treatment of numerous health problems [3,6,7,8]. Nettle is perhaps one of the most widely distributed wild plants, found in all regions of the temperate zones and growing in all seasons [3]. Although it is consumed as a green vegetable in some parts of the world, it is often considered a weed and neglected despite its high nutritional value [8,9,10,11]. All parts of the plant (flowers, stems, leaves and roots) can be used in the food, cosmetic and pharmaceutical industries for their health-promoting properties [8,12,13], as nettle is rich in various biologically active compounds and specialized metabolites (SM) [10]. The phytochemical profile of nettle leaves can be divided into several categories: terpenoids, chlorophylls and carotenoids, fatty acids, polyphenolic acids and compounds, essential amino acids, vitamins, tannins, carbohydrates, sterols, polysaccharides, isolectins and minerals [10,14,15]. According to various studies [1,2,3,16,17], the presence of chemical components in plants is variable, of inconsistent quality and depends on many abiotic, biotic and anthropological factors as well as on the conditions of raw material storage. Phytochemicals, i.e., SM are products of various metabolic pathways whose synthesis is directly influenced by these factors.

Most nettle herb is wild harvested [9], but it is challenging and even expensive to control the quality standards of wild collected plants and obtain a standardized product with a high content of SM [6]. Large differences are observed in genotype-environment interactions so that the same species has large differences in the availability of SM under different environmental conditions and habitats [18]. Therefore, cultivation is recommended, but often open field cultivation does not allow complete control over the growing conditions and consequently the SM content of the plants. There is a need for alternative systems for the production of medicinal plants, which could be an effective means to overcome the increased risk associated with open field cultivation [6,19].

Hydroponics, soilless techniques, are suitable for the cultivation of many vegetable species, but hydroponic cultivation of medicinal and aromatic plants is also attracting great interest [20,21,22]. These techniques allow for more efficient use of water and nutrients, resulting in accelerated plant growth, earlier and multiple harvests, more production cycles as crops can be grown throughout the year, and higher biomass production, ultimately leading to higher yields [21,23]. Previous research on hydroponic techniques was based on leafy vegetables with short vegetation [21,24,25,26,27,28,29], but there are few studies on hydroponic cultivation of perennial deciduous species, such as stinging nettle.

Based on the literature reviewed, which supports the fact that stinging nettle is a mesophilic, nitrophilic species and a phytoremediator that can retrovegetate during a growing season, it is our hypothesis that stinging nettle could reach its full biological potential through greenhouse cultivation using modern hydroponic techniques. Through the appropriate management of some hydroponic techniques (e.g., induction of water stress in the ebb and flow system), it is possible to create positive stress conditions that help increase SM and the nutritional value of the plant material.

Therefore, the aim of this study was to provide a detailed overview of the nutritional and functional aspects of stinging nettle, the possibilities of growing stinging nettle in the open field, and a detailed overview of hydroponic cultivation techniques for growing stinging nettle in greenhouses, such as the nutrient film technique, floating hydroponics, ebb and flow, and the aeroponic system as the best options for stinging nettle growing.

## 2. Nutritional Aspects of Nettle

Stinging nettle is a plant with great nutritional value, containing biologically significant classes of chemical compounds. Nettle leaves contain 4–7% carbohydrates, 6% dietary fiber, 0.6–9% fat, and 0.4–4% protein [30,31], while Zeipiņa et al. [32] stated that proteins from *Urtica* leaves are of better quality compared to proteins from other leafy vegetables. Among the most abundant fatty acids in nettle, palmitic acid and linoleic acid stand out [4,7,9,33,34]. It is a valuable source of vitamins A, B group vitamins, C, D, E, and K, and minerals [7,8,11,13,14,15,16,31,32,35,36,37,38], especially P, Ca, Mg, and Fe, the values of which are listed in Table 1 according to various authors.

Table 2 lists the content of some SM in stinging nettle identified in various studies. In addition to the SM listed in the table, nettle also contains essential amino acids, tannins, various flavonoids [3,7,11,13,14,34,35,37,42], volatile compounds [3,11,16,36], essential oils with carvacrol, carvone and naphthalene as major constituents [7,35,36], phytosterols, saponins [36], coumarins [14,38], amines, glucokinins [11] and terpenoids [7,8,9,12,38]. Nine carotenoids are present in stinging nettle, the most important of which are β-carotene, violaxanthin, xanthophylls, zeaxanthin, luteoxanthin, and lutein epoxide [8,10,32,38]. Nettle also contains caffeic acid, hydroxybenzoic acid, vanillic acid, coumaric acid, and quinic acid, among others, as well as many caffeic acid and quinic acid derivatives [9,12,36,42]. It is important to emphasize that the amounts of these chemical compounds vary greatly from author to author. This is because their amount depends on the origin of the plant material, the plant part, the phenological stage, and the analytical method used for their determination. Many authors emphasize the antioxidant properties of nettle, which are determined by phenols, vitamins, pigments, and other compounds [38]. The values of antioxidant capacity of nettle leaves are also listed in Table 2.

## 3. Functional Properties and Use of Nettle

Nettle, like other plants, was and is used as a general remedy, both in traditional folk medicine and in industrially manufactured products [46]. Cancer, emphysema, cirrhosis of the liver, atherosclerosis, atheroma, hypertension, neoplasms, and arthritis are all associated with oxidative damage, but foods rich in antioxidants can help the human body reduce oxidative damage from free radicals and active oxygen [16,46]. According to various studies [4,7,8,10,12,13,14,15,16,36,43,47], all parts of the nettle plant possess anti-radical and antioxidant, anti-inflammatory, antimicrobial, antifungal, antiviral, anti-ulcer, anti-rheumatic, and anti-cancer properties due to their rich phytochemical composition. The beneficial pharmacological effects of nettle leaves are probably due to the content of flavanol glycosides, phenolic and amino acids, tannins, vitamins, and carotenoids [48,49].

Fresh nettle has been used since Roman times to treat urticaria, to stimulate blood circulation and warm the joints and extremities [7,32] and to help with rheumatism and arthritis [3,14,45,46,48,49,50,51]. The aqueous and alcoholic extracts of nettle are used to treat rhinitis [9,10] and lung diseases [3,16,32,51], while it is reported that dried nettle is good for treating allergies [14,43]. It is also used as a moderate diuretic and for ailments related to urinary and renal problems [3,8,13,14,16,32,42,43,45,46,48,49], cardiovascular problems [9,10,43,45,48,52], diabetes [48], gout [8,10,13], eczema [10,32,46,48], and liver cirrhosis [16,50]. Chemotherapeutic agents from nettle are being developed for the treatment of cancer patients [32,51].

Nettle is traditionally used as a green leafy vegetable in many Mediterranean and Eastern European countries [7,9,10,14,53]. Young leaves and shoots are usually collected in spring before flowering [3,32,53], as older plants may contain higher amounts of cystolytes, which can irritate the kidneys [8]. Because of its very similar taste and texture, cooked nettle is often used as a spinach substitute [8]. It is also used in the preparation of various dishes, such as salads, soups, vegetable cakes [3,8,10,13,45,53] or rice and pasta dishes [3,53]. Very often, the herb is used to prepare tea, which can be a nutrient-rich substitute for water [8,9,10,13], in the UK it is used to prepare drinks similar to ginger beer [3,8], or it can be used for cocktails and herbal liqueurs [3,53]. Increasingly, nettle is used as livestock feed and improves the quality of meat, eggs, and dairy products [32,53]. The food industry uses nettle to produce milk curd and to extract chlorophyll (as used as green dye E140) or yellow dye from its roots. It is also used in the medical, pharmaceutical, and cosmetic industries for the production of hair shampoos and tonics [3,7,13,32,47,53]. Other traditional uses of stinging nettle include the textile industry, where it is used for light and resistant fibers from its stems [47], and the production of biopesticides [53].

## 4. Cultivation of Nettle—From Open Field to Hydroponics

Due to the high nutritional value and functional properties, as well as the wide use of stinging nettle described in Section 2 and Section 3, the demand for its fresh biomass is increasing. Nettle is traditionally collected from the wild for various purposes (organic fertilizer, food, pharmacological and cosmetic products), but this type of plant material is often of questionable quality and inconsistent chemical composition and even unsuitable for human consumption. In fact, nettle is also mentioned as a hyperaccumulating plant that has a strong tendency to collect heavy metals (Pb, Cd, Zn, As) from the soil in leaves and shoots, so wild harvested plant material may be contaminated. This ability to accumulate heavy metals and nutrients, such as nitrogen and phosphorus is sometimes used to purify the soil, which is why nettle is considered a phytoremediation plant [54,55]. Moreover, stinging nettle is a nitrophilous plant species that prefers to grow in soils rich in nitrogen and organic matter, which can lead to the accumulation of potentially harmful nitrates in plant material [40]. All this indicates the need to introduce nettle into agricultural production, but the technology of nettle cultivation is still unknown and insufficiently researched [19].

Since most research studies analyze the chemical composition of wild-collected nettles, it is important to emphasize that it is difficult to find detailed information on both appropriate cultivation techniques (field or greenhouse) and the nutrient quality or SM content of the cultivated plant material. Literature indicates that nettle can be propagated both generatively and vegetatively, by direct seeding, by seedlings, or by planting underground stems (rhizomes) [10,41]. In generative propagation by seeds, it should be noted that nettle seeds are quite small, and their absolute weight is about 0.15 g (Figure 1). Therefore, when sowing in the field, it is important to prepare the sowing layer adequately (fine preparation of the soil surface) and not to sow deeper than 1 cm. Since nettle seeds have a hard seed coat, which makes germination difficult, sowing in autumn (but before the late autumn frosts) is more favorable than sowing in spring. In this case, the dormancy of the seed is canceled, and the seeds germinate the following spring.

The distance between rows is 20 cm and it is recommended to sow about 1000 seeds/m^2^. However, it is known that seed germination is affected by numerous abiotic (humidity, heat, light, and substrate) and biotic (seed size and type, seed coat thickness and permeability, dormancy) factors, which may result in lower seed germination (20–23% under controlled conditions in the climate chamber; [56]), longer germination time, and uneven planting density. It takes a long time for seeds to germinate and some time for them to reach a favorable plant density, so weeds can also be a potential problem in field cultivation. Radman et al. [28,40] concluded that although direct seeding is a faster and cheaper method of nettle propagation, it is not used in practice because of the many limiting factors mentioned above. Vegetative propagation by rhizomes is also relatively cheap (the cost of producing and procuring planting material is eliminated), but it is a more demanding propagation method, especially for intensive production, because it requires a lot of physical labor. Rhizomes are taken from the existing planting in late fall, removing above-ground, rotten and dead parts. Then the cleaned rhizomes are separated and cut to a length of 25–30 cm. It is important to coordinate the removal with planting to keep the rhizomes out of the soil as short as possible to avoid desiccation. Another risk of this propagation technique is that the age of the rhizome and the number of dormant buds cannot be determined at the time of removal [57].

In view of all this, it is recommended to grow nettles from seedlings sown in greenhouses in early spring and planted outdoors after about 40 days, but the optimal cultivation technique is not yet well defined or researched [16,41].

Seedlings (Figure 2a) should be grown under controlled conditions in a greenhouse at a temperature of 15 or 20 °C and a relative humidity of 60%. Stepanović et al. [57] state that the optimal conditions for germination of nettle seeds are light or a 12-h light and dark period at a temperature of 15–20 °C. Temperatures of 20–25 °C are recommended for herb growth and development (Figure 2b) [30,56].

Considering that in nature nettle prefers moist sites at forest edges and along springs or rivers, the lack of precipitation during certain growth stages of the plant can be a serious problem in field cultivation [8]. Therefore, irrigation must be provided, especially in dry and semi-arid areas.

In cultivation, it is important to provide the plants with the necessary nutrients, through an appropriate and, above all, balanced fertilization. Nettle requires various nutrients for its growth and development, especially nitrogen, which is why it is considered a very nitrophilous plant species [41,57]. On the other hand, as mentioned earlier, in the cultivation of leafy vegetables, a higher nitrogen supply can lead to excessive accumulation of nitrates and negatively affect crop quality and groundwater contamination. Nitrate itself is relatively nontoxic, but its metabolites can lead to a number of health problems, such as methemoglobinemia in young children. The European Union prescribes maximum allowable levels for nitrate in lettuce and spinach in Regulation 563/2002 [58], but there are no regulations for other types of leafy vegetables [24,28]. For these reasons, the balanced application of nitrogen fertilizers is of paramount importance to enable the availability of high-quality plant material [28]. For optimum yields and high content of SM in the plant material, it is recommended to apply 100 kg/ha of nitrogen fertilizer (KAN) at several intervals during growth, depending on the amount of available nutrients in the soil and the planned number of harvests. Nettle is also rich in minerals (Table 1), such as P, Ca, Mg and Fe [28,33,40,59], which means that the soil must be supplied with these nutrients in a form in which the plant can absorb them because the uptake of nutrients by the root system of the plant depends on a number of its physicochemical properties [41].

Nettle is characterized by its ability to retrovegetate, so it can be harvested several times during the growing season (Figure 3). The stage at which it is harvested depends on the objective of cultivation, e.g., intensive growth of the plant before flowering when the objective of cultivation is a fresh leaf for human consumption. Harvesting is carried out above the first two nodules to allow the plant to regenerate [9,41].

## 5. Possibilities and Challenges of Hydroponic Nettle Cultivation

In general, the quality of plants grown outdoors varies from year to year due to fluctuating environmental conditions (early or late frost, hail, extreme temperature changes) and it is difficult or even impossible to obtain fully standardized plant material [19].

Cultivation under controlled conditions in a greenhouse provides the opportunity to increase and equalize the quality of the raw material by monitoring and managing abiotic factors during plant growth [21]. New strategies and cultivation technologies are constantly being developed to address specific constraints during production and, more importantly, to reduce the negative impact of environmental factors (climate-related floods, wildfires, drought, and heatwaves, and in some cases, hurricanes) and adapt to new market demands and consumer needs [30].

Hydroponic cultivation techniques, i.e., cultivation in nutrient solution with or without substrates, is a successful alternative to traditional agriculture. Hydroponic cultivation eliminates potential problems caused by soil contamination with heavy metals and pesticide residues or by complex soil-nutrient relationships, allows lower water consumption, and precise and balanced plant nutrition according to the needs of the cultivated species to ensure high yields and nutrient-rich raw material [23,60,61,62].

Hydroponics takes advantage of the fact that plants can synthesize all necessary metabolic products from inorganic ions, water and CO_2_ using solar energy. All nutrients are provided in their inorganic form by the water solution according to the average requirement of a particular crop during cultivation, but some organic compounds, such as iron chelates, may also be present [63]. Hydroponic cultivation can have high efficiency in water use and fertilization and low environmental impact [24]. According to Nicola et al. [21] and Resh [64], hydroponic systems allow the reuse of the nutrient solution through recirculation, where the spent solution is collected and returned to the system after sterilization. This system is called closed and is suitable for environmentally sensitive areas to protect soil and groundwater. If the nutrient solution is used only once, it is an open system. In an open system, environmental impacts can also be reduced by minimizing water and nutrient losses through the use of sensing technologies and nutrient solution delivery strategies, with the goal of minimizing solution runoff from the system [65].

However, hydroponic techniques require a larger initial investment in greenhouse and equipment, but on the other hand have long-term and multiple benefits. For successful modern hydroponic production of nettles, it is very important to have appropriate tools and equipment to adjust the vegetation factors (light, heat, humidity and composition) to the optimal value.

Light is a vegetation factor needed for the basic process in plants—photosynthesis. Cultivation under controlled conditions allows plant material to be available in the market throughout the year, for example, in winter when sunlight is absent or when the change in the spectrum causes undesirable stretching of plants, resulting in uneven growth [66]. Therefore, additional lighting must be used during a part of the year when there is not enough light. Supplemental lighting in the form of LED diodes is common practice. Different wavelengths of LEDs or their combination can increase the SM content (vitamin C, anthocyanins, and total phenols) of leafy vegetables, and thus also in nettles. However, this is species-specific, so the combinations and duration of light treatments for each species need further research [67]. Although moderate light treatments can have a positive effect on the content of the plant SM, on the other hand, an excessive amount of light is associated with a reduction in the amount of chlorophyll. Therefore, it is common practice to install shade curtains in greenhouses to protect plants during seasons with strong sunlight. However, photoselective nets of different colors and shading efficiencies can also be used to stimulate SM production [68]. For example, stinging nettle is a mesophilic species that, according to its biological requirements, does not have a great need for light and heat, and therefore, requires shaded places without direct sunlight [57].

Because of the greenhouse effect in the warm months (June to September in temperate climates), ventilation is required. Usually, there is an automatic system controlled by a computer. When a certain temperature is reached, the sidewalls and part of the roof are opened to allow the warmer air to leave the greenhouse with the help of fans. In the colder months, the shade curtain is often used as an energy curtain along with a heater to minimize heat loss. The heater is accompanied by fans that drive warm air through the greenhouse to distribute heat more evenly.

The ability to hold water vapor in the air is determined by the air temperature. The proper balance of humidity and temperature in the greenhouse is critical to the health and success of plant production. This is usually balanced by a ventilation system, sometimes in combination with misting [69].

The composition of the air in the greenhouse is also very important; carbon dioxide (CO_2_) is necessary for plant photosynthesis. On a cold, sunny day when the ventilation is not open, the concentration in the greenhouse can drop to only half of that outside. The atmosphere contains about 340 ppm (0.03%) CO_2_. Increasing CO_2_ by 3 to 4 times accelerates photosynthesis, but should be performed with caution, as plant organs can be damaged if there is not enough light, heat, water, and plant nutrients. Pure CO_2_ (bottled, compressed, dry ice) is mainly used for research purposes [70].

Depending on the hydroponic technique to be used, the greenhouse must be equipped with a system of tanks, pumps, and pipes for preparing and transporting the nutrient solution to the plants (Figure 4).

### 5.1. Nutrient Solution Management

In hydroponic cultivation with water, nutrients and salt ions are needed to produce the nutrient solution. The advantage of nutrient salts is that they are highly pure chemical compounds consisting of two to three elements, i.e., nutrients. All nutrient solutions used in hydroponic growing are essentially derived from the original protocol developed by Hoagland and Arnon in 1938. A standardized solution consists of the following macronutrients: nitrogen (N), potassium (K), phosphorus (P), calcium (Ca), magnesium (Mg) and sulfur (S); and trace elements: the soluble form of iron (Fe), boron (B), copper (Cu), manganese (Mn), zinc (Zn), molybdenum (Mo) and chlorine (Cl). Sometimes silicon (Si) and selenium (Se) are also added to the solution, although they are not biogenic elements. However, they are considered useful for plants because they increase stress tolerance, vegetative growth, and seed production [71]. It is also important to emphasize that the availability of certain biogenic elements, especially nitrogen, can negatively affect certain SMs, such as polyphenolic compounds [40], so growing SM rich plant material is very often a trade-off between a high yield and the content of SM in the harvested product [9].

According to Sonneveld and Voogt [72], it is necessary to perform a chemical analysis of the water to be used before preparing the nutrient solution. For the preparation of the nutrient solution, fewer salts are added to the solution, which contains ions that are also present in the water (most commonly Ca^2+^, Mg^2+^ and SO_4_^2−^). When a significant concentration of these ions is present, the pH of the nutrient solution easily becomes too high due to the alkaline buffering capacity of the carbonate. Regardless of the hydroponic technique, the finished nutrient solution is made from 100-fold concentrated solutions, based on the concentration of a solution supplied to each plant through a fertigation system. Therefore, in each hydroponic production, there are at least three containers for the concentrated solution (Figure 4).

The salts are prepared in two concentrated base solutions in two separate containers (A and B). Because of the chemical reactions and the possible sinking of the compounds formed, the calcium salts must be separated from the sulfates and phosphates. The third container I contains a solution of nitric or phosphoric acid, which serves to regulate the pH of the solution (Figure 3). Although the HCO_3_ ion is not a plant nutrient, it must be considered in the preparation of the nutrient solution. Its accumulation greatly increases the pH of the solution, which can have a negative effect on nutrient uptake. The availability of P and Mn is strongly influenced by pH. At values > 6.5, the concentration may be low regardless of the concentration supplied with irrigation, because these elements are less soluble at high pH values. Therefore, acids are used to neutralize it and achieve the desired pH [72,73].

Plant species grown using hydroponic techniques can differ significantly in the uptake of individual nutrients, which is influenced by many abiotic (temperature and humidity, pH, oxygen content of the nutrient solution, amount of light, and CO_2_ concentration) and biotic factors (stage of growth and development, presence of harmful organisms). Therefore, during cultivation, it is necessary to monitor the above parameters of the nutrient solution on a daily basis and correct them if necessary [74]. The same assertion is made by Tomasi et al. [60], pointing out that the management of the cultivation conditions and, in particular, the concentration of the nutrient solution is one of the most important aspects for successful hydroponic production. The same authors note that electrical conductivity (EC, dS/m), the chemical form of the elements (e.g., in hydroponics, N is usually supplied as NO^3−^ because the NH_4_^+^ form is not immediately available to the plant), temperature, and pH of the solution can affect growth, quality, and plant health, so it is necessary to monitor and correct the solution throughout the growing season. EC indicates the amount of soluble salts in the solution, and optimal values can vary widely among different crops [75].

The optimum pH of the final nutrient solution for hydroponic vegetable species ranges from 5.8 to 6.5, the optimum EC values of the nutrient solution are 1.5–3 dS/m, while the optimum amount of dissolved oxygen in the nutrient solution is 2–4 mg/L [76]. According to Nguyen et al. [63], plant roots require a constant supply of oxygen. If the roots become anoxic, they can no longer absorb nutrients and transport. In hydroponics, there are different solutions for this depending on the growing technique, but in cases where the roots are in constant contact with the nutrient solution, there is usually a pump system that oxygenates the solution to a satisfactory level (above 1 mg/L).

In modern hydroponic growing systems, the amount of oxygen, EC and the pH of the solution, as well as the dilution of the concentrated solution, are automatically controlled by a computer system that uses special sensors. The software sets the target pH and EC of the nutrient solution. The device for mixing concentrated solutions with water measures the set factors of the diluted solution and independently corrects the volumes of concentrated solutions (A and B) and acI(C) until the set values are reached. In floating hydroponics, there are also probes that are immersed in the nutrient solution in basins with plants. The data is transmitted in real time to the cloud, from where it can be read at any time via a mobile app or computer. In this way, a faster response is possible when the parameters of the nutrient solution need to be corrected, which undoubtedly has a positive effect on the success of the cultivation [77].

It can be concluded that the nutrient solItion is certainly one of thI most important factors in the quality and yield of the plant material, as it allows direct control of the amount of each biogenic element, avoids nutrient antagonism, and ensures optimal plant nutrition [78]. To obtain a high content of SM and an adequate yield, it is necessary to determine for each species the optimal composition of the nutrient solution, the seeding density and the appropriate assortment depending on the growing period and the intended use of the final product [72]. To date, a few studies [6,17] have been conducted on nutrient solutions suitable for hydroponic nettle cultivation.

### 5.2. Suitable Hydroponics Techniques for Nettle Cultivation

In general, hydroponic systems for the production of green leafy vegetables can be divided into two groups according to Resh [64]. Hydroponic systems in which the root of the plant is submerged, i.e., in constant contact with the nutrient solution (nutrient film technique, floating hydroponics and aeroponic system). Systems in which the inert substrates in which the plants are rooted are soaked at intervals (ebb and flow system). The following text is a brief description of the rare studies on hydroponic stinging nettle cultivation and the factors tested, which provide valuable results and the basis for future research.

Most research to date has been conducted using the floating hydroponic system. In floating hydroponics, plants are grown on an aerated nutrient solution in tanks 20 to 25 cm deep (Figure 5). Plants are grown in Styrofoam containers or boards that float on the nutrient solution. This is an ideal technique for growing leafy vegetables, such as lettuce, gentian, dill, arugula, radicchio, spinach, and basil, which features a shorter production cycle and higher growing density [21,60].

Radman et al. [6] studied the effects of three seeding densities (0.2, 0.5 and 0.9 g/m^2^) and two substrates (perlite, vermiculite) on nettle yield and number of harvests. Floating hydroponics proved to be a good cultivation technique for nettles. High yields and a greater number of harvests were obtained in the months when the nettle rested in the open field. The highest yields (1.41 kg/m^2^ and 1.22 kg/m^2^) were obtained in the spring growing season in combination with perlite × 0.2 g/m^2^ and vermiculite × 0.2 g/m^2^ seeding density.

The objective of the study by Radman et al. [17] was to determine the effect of two standard nutrient solutions for leafy vegetables on the yield and mineral content of nettle in multiple harvests (Figure 5a–d). The first nutrient solution was prepared according to Tesi with slightly lower EC values (2.3 dS/m) and the second according to Osvald (2.5 dS/m). Both nutrient solutions proved to be adequate for growing nettles in the floating system, but the Tesi solution had higher values for most of the observed traits (dry matter, N, P, K, Fe).

In the aeroponic system, plants are placed in lattice pots in holes in Styrofoam panels placed horizontally or at an angle of 45 to 60 degrees (A-frames). The roots of the plants hang in a closed chamber and are exposed to the air. The nutrient solution is sprayed every few minutes for about 15 s so that the root system is in constant contact with it [64,79]. In the study by Hayden [23], stinging nettle was grown in an aeroponic A-frame system, with no information on the nutrient solution used. The control group was grown in a soilless potting mix. The aeroponic nettle produced the same biomass in the air, but lower biomass on the ground compared to the control group.

In a study by Pagliarulo et al. [20], two nutrient solutions modified from Resh (1998) with different phosphorus and potassium contents were tested on nettles grown in hydroponic systems (aeroponic and soilless systems in controlled environments). The change in PK ratio had no significant effect on yield, but the soilless medium treatments produced greater root biomass and the same shoot biomass compared to aeroponics.

Most research has addressed the cultivation of nettles in floating hydroponics [6,17] and in aeroponic systems [20,23], but considering the techniques and nettle as a species, it shows great potential for cultivation with both nutrient films and ebb and flow techniques. The potential of cultivation with the aforementioned techniques comes from the rational use of nutrients, which results in less nitrate accumulation, and the ability of stinging nettle to retrovegetate, which allows for multiple harvests. In addition, the controlled management and control of stressors through the use of these techniques (especially in the ebb and flow system) can influence the increase of specialized metabolites.

In the NFT nutrient film technique, an aerated nutrient solution up to 1 cm high flows continuously in shallow channels containing growing containers with a relatively small amount of substrate and plants. Because the solution is constantly in motion, the ion concentration in the root zone does not increase. NFT maximizes water use efficiency by recycling a nutrient solution that is sterilized before being returned to the system [64,80].

The ebb and flow system (a”so c’lled flood and drain) works on the principle of the time intervals between the availability of nutrient solutions and dry periods (water stress) to which plants are exposed. The benches on which the containers with the plants are placed are soaked with a nutrient solution. After a certain time, interval (programmed according to the plant grown), the nutrient solution drains from the bench. The system is closed, and the solution is recycled [23,64]. The use of ebb and flow systems and the regulation of the time interval between two irrigations with nutrient solution (which cause water stress) can affect the morphological and nutritional characteristics of plants [81]. Water stress can have a positive or negative effect on the chemical composition of plants, depending mainly on the length of the stress period, the genetic characteristics of the plant material, and the phenological phase of the plant. Water stress was shown to have a negative effect on the uptake of phosphorus and calcium in arugula and of phosphorus and magnesium in spinach. On the other hand, water stress had a positive effect on the content of bioactive compounds: vitamin C, total phenols, non-flavonoids and glucosinolates [82].

As you can see from the described chapter, research on hydroponic stinging nettle cultivation is scarce, but it provides valuable guidance for the selection of appropriate hydroponic techniques and also for their appropriate management to induce positive stress in stinging nettle cultivation, which increases the content of SM.

### 5.3. Management of Hydroponic Techniques Affecting SM in Plants

The induction of positive stress by the cultivation method and the application of agrotechnical measures at different stages of plant development can influence the content of SM [82]. For example, Šic Žlabur et al. [66] proved that induced mechanical stress by brushing is a good method for growing basil, the application of which provides high quality plant material with high nutritional potential and a significantly higher content of antioxidants and phytochemicals important for human health. Hydroponic growing techniques also offer the possibility of positively affecting stress in plants in various ways, which can lead to plant material with a richer SM composition and thus higher nutrient quality [68]. Indeed, in plants exposed to stress, the total content of flavonoids, phenols and polyphenols, vitamins C and E, carotenoids, and antioxidant enzymes, i.e., all compounds of secondary metabolism responsible for the defense mechanism of plants, increases [66,83,84].

According to Hayden [23] and Maggini et al. [19], it is possible to improve the synthesis and accumulation of SM in medicinal plants by manipulating the nutrient solution for certain SM provoking stress conditions. According to Maggini et al. [19], the content of basil SM, rosmarinic acid, can be increased when plants are exposed to moderate NaCl salt stress, moderate hypoxia conditions, or altered nitrogen supply. More specifically, when the nutrient solution was manipulated with NO_3_^−^ concentration, it was found that rosmarinic acid content was higher when NO_3_^−^ was decreased. In general, plants SM, such as β-carotene, vitamins, flavanols, lycopene and phenols are stimulated under N deficiency conditions. This can be very interesting for controlling the amount of nitrate in the plant material by changing the concentration and composition of the nutrient solution to improve the SM content while reducing the accumulation of undesirable compounds [68]. Leafy vegetables and nettles tend to accumulate nitrate, so reducing nitrate levels is an important aspect of vegetable production. Nitrate accumulation in plants is influenced by many factors, including fertilization, soil properties, growth stage, air temperature, light intensity, and harvest time [21]. The same authors state that nitrate content in leafy vegetables grown in hydroponics can be reduced by stopping fertilization a few days before harvest. Radman et al. [85] also indicate the possibility of reducing nitrate content in lamb’s lettuce (*Valerianella locusta* L.) by replacing the nutrient solution with water 3 days before harvest. However, the authors indicate that this process resulted in a lower amount of dry matter and minerals in the plant material. Gonnella et al. [24] state that replacing the nutrient solution with water 3 days before harvest of lamb’s lettuce had no effect on yield and organoleptic characteristics.

Increasing the potassium or magnesium content in the nutrient solution can also have a positive effect on the SM content of the plants. However, it is advisable to handle this with caution, as nutrient antagonism may occur [68].

The ebb-and-flow method is best suited for studying the effects of water stress on SM content in plants. The dry period, i.e., lack of water (water/drought stress), significantly affects the content and profile of SM. During this period, the plant begins to accumulate a greater amount of SM in response to its defense mechanism [86]. According to Yadav et al. [84], De Abreu and Mazzafera [87], Selmar and Kleinwächter [88], and Bloem et al. [89], it is possible to increase the content of SM in medicinal and aromatic plants by causing moderate water stress during cultivation. It is of great importance to set an appropriate time interval for irrigation so that plants are sufficiently stressed to produce larger amounts of SM, but this stress does not lead to wilting or irreversible damage to plant tissues.

Since even moderate stress can cause yield loss, it is important in agricultural production to find the right balance between generated stress, which gives the plant material a richer SM content, and a satisfactory yield [88,90].

## 6. Conclusions

Considering the peculiarities of nettle as a wild plant, such as the possibility of re-vegetation, the tendency to accumulate nitrates and heavy metals in the soil, resulting in a variable composition of various specialized metabolites, it is necessary to control the nutritional properties of this plant material. Cultivation allows easier control of chemical properties, which is a necessity in the case of stinging nettle. Cultivation of stinging nettle in the open field proves to be a suitable option. However, as it is not possible to fully control some important abiotic factors, such as temperature, light and humidity, this method of cultivation faces many challenges, especially in the face of increasingly volatile climate change and the increasing occurrence of extreme weather events. The cultivation of nettles using modern hydroponic techniques in greenhouses offers itself as a suitable solution to overcome the above challenges. The literature review has revealed numerous advantages of using certain techniques of modern hydroponic cultivation for stinging nettle. These include the ability to control and manage abiotic factors for both the greenhouse and the nutrient solution, balanced and rational fertilization, optimized water management and supply, and most importantly, the production of plant material with adequate yields and high levels of specialized metabolites. With the ebb and flow system, it is even possible to induce water stress, the optimized application of which can be successfully used to stimulate the synthesis and accumulation of secondary plant compounds, such as polyphenols. After all, the floating hydroponics and the ebb and flow system prove to be optimal hydroponic techniques for greenhouse cultivation of nettles, ultimately resulting in uniform plant material with consistent chemical composition, high yields and a high content of specific metabolites.

## Figures and Tables

**Figure 1 plants-11-00483-f001:**
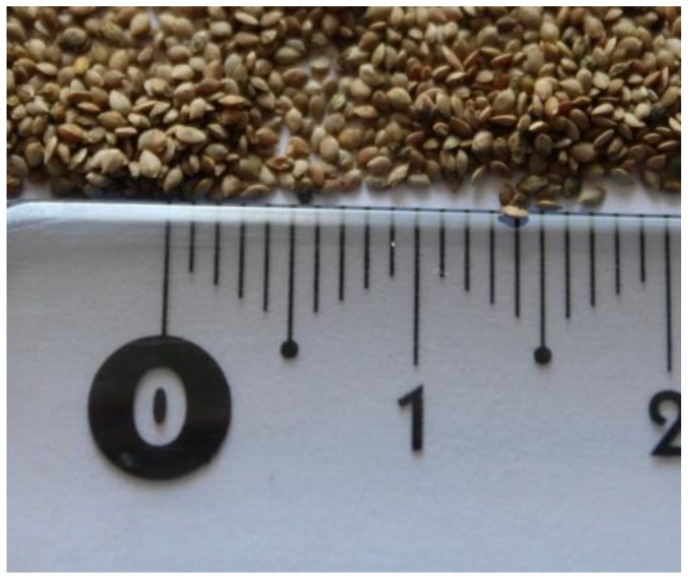
*Urtica dioica* seeds.

**Figure 2 plants-11-00483-f002:**
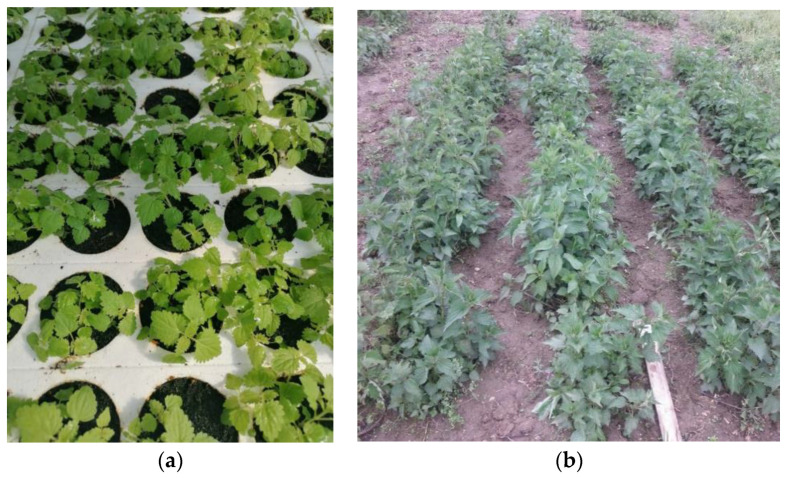
(**a**) Nettle seedlings in polystyrene containers; (**b**) Nettle herb growing in the open field.

**Figure 3 plants-11-00483-f003:**
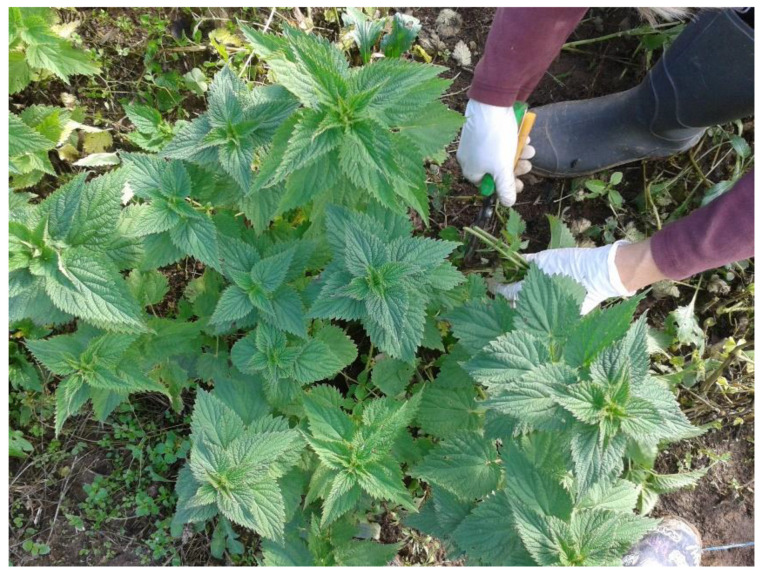
Harvesting nettle in open field.

**Figure 4 plants-11-00483-f004:**
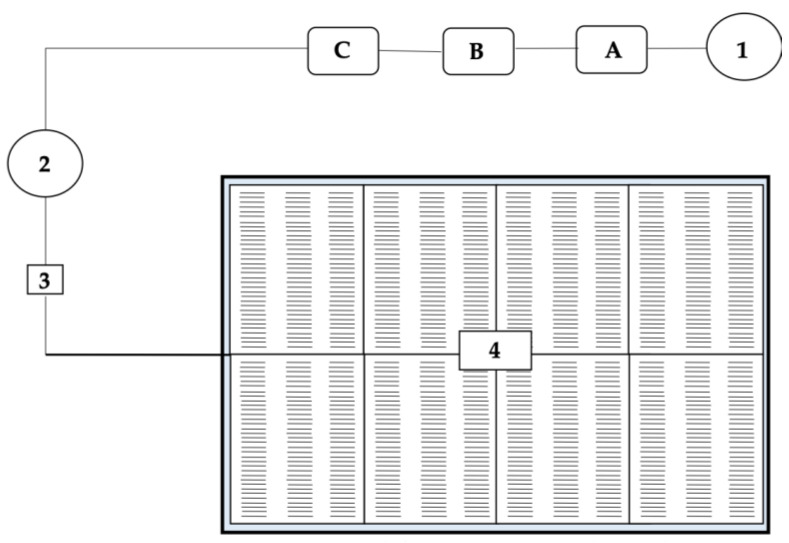
Graphical scheme of floating hydroponics cultivation. 1—water; A, B, C—tanks for concentrated nutrient solutions and injectors; 2—tank for standard nutrient solution; 3—pump; 4—basin with polystyrene containers or boards.

**Figure 5 plants-11-00483-f005:**
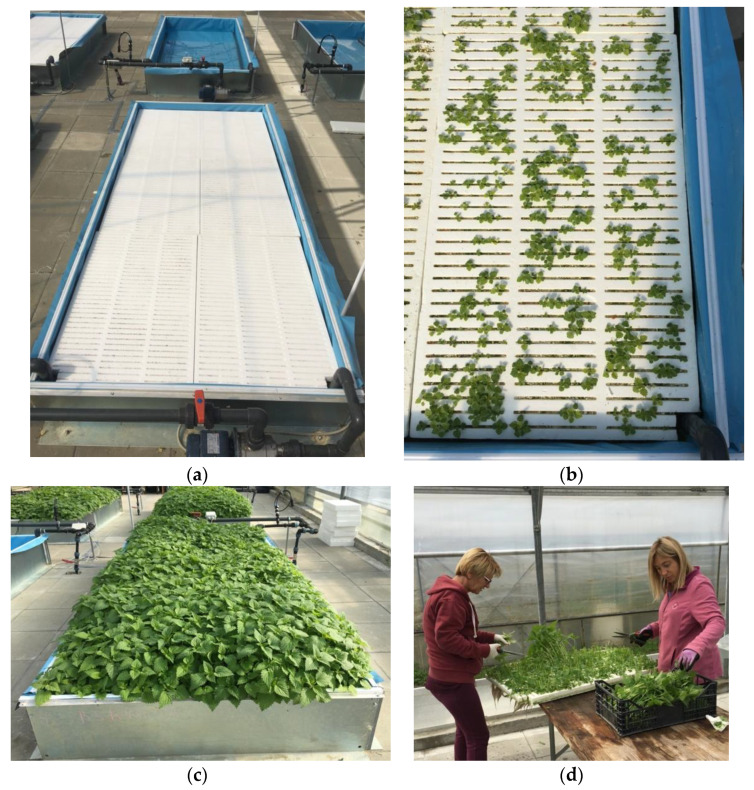
(**a**) Placing of polystyrene boards in floating hydroponics basins after initial germination of nettle seeds (**b**) Beginning of herb growth; (**c**) Nettle plants before flowering, ready for harvest; (**d**) Harvest above plants first two nodules to allow retrovegetation.

**Table 1 plants-11-00483-t001:** The content of the main minerals in nettle depends on the habitats and the part of the plant analyzed.

Mineral	Content	Method Used	Cultivation or Wild Harvest Location	Plant Part	Reference
Ca	28,605 μg/g	US EPA, 1994	wild harvested (Serbia)	dried leaves	[7]
	853–1050 mg/100 g	unknown	unknown	whole plant	[9]
	3.43%	Nowosielski, 1988	open field (Poland)	leaves	[16]
	2.15–3.09%	AOAC, 1995	floating hydroponics (Croatia)	leaves	[17]
	168.77 mg/100 g	AOAC, 2005	wild harvested (Nepal)	dry leaves powder	[31]
	278–788 mg/100 g	AOAC, 2000	open field (USA)	shoots before flowering	[33]
	323 mg/100 g	AACC Int., 2000	open field (South Africa)	leaves	[37]
	3.04%	71/250/EEC	open field (Lithuania)	leaves	[38]
	2.63–5.09%	AOAC, 1995	wild harvested (Macedonia)	leaves	[39]
	5.21%	AOAC, 1995	open field (Croatia)	leaves before flowering	[40]
Fe	150.97 μg/g	US EPA, 1994	wild harvested (Serbia)	dried leaves	[7]
	227.89 mg/100 g	unknown	unknown	leaf powder	[8]
	2–200 mg/100 g	unknown	unknown	whole plant	[9]
	79.20–89.50 mg/kg	AOAC, 1995	floating hydroponics (Croatia)	leaves	[17]
	886–3651 mg/kg	AOAC, 1995	open field (Croatia)	leaves	[28]
	227.89 mg/100 g	AOAC, 2005	wild harvested (Nepal)	dry leaves powder	[31]
	1.2–3.4 mg/100 g	AOAC, 2000	open field (USA)	shoots before flowering	[33]
	2.5 mg/100 g	AACC Int., 2000	open field (South Africa)	leaves	[37]
	224.78 mg/kg	73/46/EEC	open field, (Lithuania)	leaves	[38]
	2765 ppm	AOAC, 1995	open field (Croatia)	leaves before flowering	[40]
	145–2717 mg/kg	AOAC, 1995	open field (Croatia)	leaves before flowering	[41]
Mg	8699.76 μg/g	US EPA, 1994	wild harvested (Serbia)	dried leaves	[7]
	0.34%	Nowosielski, 1988	open field (Poland)	leaves	[16]
	0.23–0.48%	AOAC, 1995	floating hydroponics (Croatia)	leaves	[17]
	104 mg/100 g	AACC Int., 2000	open field (South Africa)	leaves	[37]
	0.61%	73/46/EEC	open field, (Lithuania)	leaves	[38]
	2.51–3.56%	AOAC, 1995	wild harvested (Macedonia)	leaves	[39]
P	50–265 mg/100 g	unknown	unknown	whole plant	[9]
	0.39%	Nowosielski, 1988	open field (Poland)	leaves	[16]
	0.41–0.49%	AOAC, 1995	floating hydroponics (Croatia)	leaves	[17]
	82.6 mg/100 g	AACC Int., 2000	open field (South Africa)	leaves	[37]
	0.82%	71/393/EEC	open field (Lithuania)	leaves	[38]

**Table 2 plants-11-00483-t002:** The content of SM and antioxidant capacity in nettle depends on the habitats and the part of the plant analyzed.

Specialized Metabolite	Content	Method Used	Cultivation or Wild Harvest Location	Plant Part	Reference
Total carotenoids	1.62 mg/g	spectrophotometry	wild harvested (Latvia)	leaves (ethanol extract)	[1]
	51.4–74.8 μg/g	Wills et al., 1988	wild harvested (Spain)	leaves	[4]
	5.47 mg/g	Wellburn, 1994; Dere et al., 1998; Pavlić et al., 2016	wild harvested (Serbia)	dried leaves (96% ethanol extract)	[7]
	2.95–8 mg/100 g	unknown	unknown	whole plant	[9]
	33.03 mg/100 g	Castro– Puyana et al. (2017)	wild harvested (Croatia)	leaves and stalks	[13]
	1.31 mg/g	Rumiñska et al., 1985	open field (Poland)	leaves	[16]
	3496.67 μg/g, db	Ranganna (2001)	wild harvested (Nepal)	dry leaves powder	[31]
	0.55 mg/g	spectrophotometry	wild harvested (Latvia)	shoots	[32]
	0.216–0.323 mg/g	Holm, 1954 and Van Wattstein, 1957	wild harvested (Bosnia and Herzegovina)	leaves	[35]
	15.36 mg/100 g	Strumite et al., 2015	open field (Lithuania)	leaves	[38]
	0.81–1.01 mg/g	Porra et al., 1989	open field (Poland)	leaves	[43]
β-carotene	3.8–5.6 μg/g	Wills et al., 1988	wild harvested (Spain)	leaves	[4]
	5035–7860 IU/100 g	colorimetry	open field (USA)	shoots before flowering	[33]
	58,059 μg/100 g	colorimetry	open field (South Africa)	leaves	[37]
Total chlorophyll	24.13 mg/g	Wellburn, 1994; Dere et al., 1998; Pavlić et al., 2016	wild harvested (Serbia)	dried leaves (96% ethanol extract)	[7]
	4.8 mg/g	unknown	wild harvested	leaves	[8]
	611.19 mg/100 g	Castro–Puyana et al. (2017)	wild harvested (Croatia)	leaves and stalks	[13]
	9.66 mg/g	Rumiñska et al., 1985	open field (Poland)	leaves	[16]
	1.02–1.174 mg/g	Holm, 1954 and Van Wattstein, 1957	wild harvested (Bosnia and Herzegovina)	leaves	[35]
	2.17 mg/g	spectrophotometry	wild harvested (Latvia)	shoots	[32]
	8.03–9.45 mg/g	Porra et al., 1989	open field (Poland)	leaves	[43]
Chlorophyll a	5.56 mg/g	spectrophotometer	wild harvested (Latvia)	leaves (ethanol extract)	[1]
	16.55 mg/g	Wellburn, 1994; Dere et al., 1998; Pavlić et al., 2016	wild harvested (Serbia)	dried leaves (96% ethanol extract)	[7]
	0.698–0.882 mg/g	Holm, 1954 and Van Wattstein, 1957	wild harvested (Bosnia and Herzegovina)	leaves	[35]
	67.29 mg/100 g	Strumite et al., 2015	open field (Lithuania)	leaves	[38]
Chlorophyll b	1.84 mg/100 g	spectrophotometer	wild harvested (Latvia)	leaves (ethanol extract)	[1]
	7.58 mg/g	Wellburn, 1994; Dere et al., 1998; Pavlić et al., 2016	wild harvested (Serbia)	dried leaves (96% ethanol extract)	[7]
	0.285–0.320 mg/g	Holm, 1954 and Van Wattstein, 1957	wild harvested (Bosnia and Herzegovina)	leaves	[35]
	29.14 mg/100 g	Strumite et al., 2015	open field (Lithuania)	leaves	[38]
Total phenolics	128.75 mg GAE/g	unknown	unknown	leaf powder	[8]
	380.90 mg/100 g	Repajić et al., 2020	wild harvested (Croatia)	leaves and stalks	[13]
	14.47 mg/g	Slinghart et al., 1977	open field (Poland)	leaves	[16]
	140 mg GAE/g	Folin–Ciocalteu	wild harvested (Italy)	leaves	[12]
	450.81–539.27 mg GAE/g	Folin–Ciocalteu	unknown (Serbia)	dried leaves (different extraction methods)	[14]
	128.75 mg GAE/g	Ranganna, 2001	wild harvested (Nepal)	dry leaves powder	[31]
	26.78 mg GAE/g	unknown	wild harvested (Turkey)	USB extract	[34]
	208.37 mg GAE/g	Folin–Ciocalteu	wild harvested (Bosnia and Herzegovina)	leaves	[35]
	118.4 mg GAE/g	Folin–Ciocalteu	open field (South Africa)	leaves	[37]
	8.87 mg GAE/g	Folin–Ciocalteu	open field (Lithuania)	leaves	[38]
	732.49 mg GAE/100 g	Ough and Amerine, 1988	open field (Croatia)	leaves before flowering	[40]
	22.01–24.94 mg/g	Folin–Ciocalteu	open field (Poland)	leaves	[43]
	7.9 g/100 g	Folin–Ciocalteu	wild harvested (Portugal)	dry aerial parts during flowering	[44]
	28.42 μg/g	Orčić et al., 2014	wild harvested (Serbia)	herb	[45]
Vitamin C	20–60 mg/100 g	unknown	unknown	whole plant	[9]
	8.4 mg/g	Kampfenkel et al., 1995	wild harvested (Italy)	leaves	[12]
	0.5–1.1 mg/100 g	AOAC, 2000	open field (USA	shoots before flowering	[33]
	14.2 mg/100 g	HPLC	open field (South Africa)	leaves	[37]
	8.53 mg/100 g	Latimer, 2016	open field (Lithuania)	leaves	[38]
	63.75 mg/100 g	AOAC, 2002	open field (Croatia)	leaves before flowering	[40]
Antioxidant capacity	60 mg TEAC/g	Brand-Williams et al., 1995	wild harvested (Italy)	leaves	[12]
	26.5 μM Trolox/g	Re et al., 1999	open field (Poland)	leaves	[16]
	66.3% DPPH	Nuengchamnong et al., 2009	wild harvested (Nepal)	dry leaves powder	[31]
	0.85% DPPH	DPPH	wild harvested (Bosnia and Herzegovina)	leaves (ethanol extract)	[35]
	65.1% DPPH	Brand-Williams et al., 1995	open field (South Africa)	leaves	[37]
	70.37% DPPH	Zeipina et al., 2015	open field (Lithuania)	leaves	[38]
	1936.58 mM Trolox/L	Miller et al., 1993; Re et al., 1999	open field (Croatia)	leaves before flowering	[40]
	10.95–11.80 μM Trolox/g	Re et al., 1999	open field (Poland)	leaves	[43]

## Data Availability

Not applicable.

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
