# Peer review of "Nettle Cultivation Practices—From Open Field to Modern Hydroponics: A Case Study of Specialized Metabolites"

_plants, 2022, doi:10.3390/plants11040483_

Round 1

Reviewer 1 Report

This review is not coherent with its title. Hydroponic techniques are only a part of the manuscript: the first part is devoted to listing the components called specialized metabolites present in nettle together with a confusing list of the potential biological activities of this edible plant. Hydroponic cultivation is only a limited part of the manuscript and above all, this technique is well-known and applied to many horticultural cultivations. Maybe for nettle is not so applied and in fact, the authors declare few published articles regarding nettle but then why do a review dedicated to this issue?

Moreover, there are some superficialities in this manuscript for example in the chemical part:

what is the meaning of specialized metabolites? The list of them reported by the authors at lines 41-43 represents more or less all kinds of plant metabolites, therefore are there also nonspecialized metabolites in a plant?  and what is the difference between phytochemicals and specialized metabolites reported at line 46?

The reported value of 66% of the protein in the nettle leaves related to ref 32 is rather strange: I suggest the author verify better this information

The list of the different metabolites found in nettle reports some imprecision for example Isorhamnetin is a flavonoid.

Table 2 reports the so-defined specialized metabolites but are the fatty acids specialized metabolites?  What is the difference between the polyphenols and the total phenolics? Why in table 2 under the column of specialized metabolites, the antioxidant activity has been added?

Author Response

Dear respectable Reviewer,

Thank you for your valuable suggestions. We have reviewed the spelling and also checked the entire text for English language and style and made some corrections.

We have also used the "Track Changes" option and by yellow colour we highlighted the parts of the text we have corrected according to your valuable suggestions. The major structural changes to the text (restructuring of paragraphs) were made in chapter 5, which we did not mark in yellow.

Reviewer 2 Report

The authors reviewed modern hydroponic techniques to cultivate nettle to produce health-promoting compounds in it. It covers major considerations of nettle cultivation and health-promoting compounds that were previously reported and studied. It is really nice the authors list previous publications with information in the table which is great for potential readers. In addition, the figures and schemes to explain the hydroponics cultivation system helped me to understand. 

I do not see any problem in research review contents, in-depth review, and wide range of content coverage from cultivation to nutrient value. I think it is possible to publish as it is. 

Author Response

(The authors gave the same response as above.)

Reviewer 3 Report

Dear Authors,
The work presented for evaluation, entitled: "Modern Hydroponic Techniques of Nettle Cultivation - A Case Study of Specialized Metabolites" is an interestingly described case study
and deserves to be published in your journal. However, the authors made a few mistakes that should be eliminated before going to print.
Here are the detailed notes:
1) the abstract is treated in a general manner and does not contain all the elements of the correct abstract, i.e., it contains too long an introduction, the purpose of the study is not very clear, there is no methodology of the research carried out and the most important conclusions are missing.
2) The essence of scientific cognition and research processes are included in the content of the work. The introduction of the work should clearly define the goal in the final paragraph and should present an alternative research hypothesis to the null hypothesis, to verify it later in the work.
3) Work is not clearly structured. There must be a certain sequence in the research process, reliability and a clear arrangement of theses ensuring the proper structure of the content of the work. The introduction merges with the rest of the work and does not contain a methodological part, where tips should be provided on the implementation of the
concept of a scientific work, how literature was collected for the work, what databases were used. Research methods, techniques, and tools, including optimal research activities, should also be presented in this chapter.
4) In the description of nettle cultivation methods, field cultivation was basically neglected, which the authors could refer to as a control object and compare the advantages and disadvantages of the discussed soilless cultivation systems against the background of field cultivation.
5) The title of the work "Modern Hydroponic Techniques of Nettle Cultivation - A Case Study of Specialized Metabolites" obliges us to do something. There is no separate subchapter on the influence of cultivation systems on the content of bioactive compounds in nettle plants in the study.
6) The conclusions of the work should be more condensed and contain summary statements, including indications of practical applications.

Author Response

(The authors gave the same response as above.)

Round 2

Reviewer 1 Report

The manuscript has been reviewed by the authors and now is acceptable for publication in Plants. 

Please check again English by a mother tongue expert

Author Response

Respected, we send you the final version of the paper after the second revision.

Comments from reviewer 1:

The manuscript has been reviewed by the authors and is now suitable for publication in Plants. Please have the manuscript reviewed again by a native English speaking expert.

RESPONSE: Dear reviewer, thank you for your valuable suggestions. We have checked the spelling and also double check the whole text for English language and style and made some corrections.

Thank you for your valuable time and suggestions!

Best regards, in front of the Authors' team.

                                                                                               Sanja Radman
